# Effect Sizes for Estimating Differential Item Functioning Influence at the Test Level

**W. Holmes Finch** [1,*] and **Brian F. French** [2]

[1] Department of Educational Psychology, Ball State University, Muncie, IN 47304, USA

[2] Learning and Performance Research Centre, Psychometric Laboratory, Washington State University, Pullman, WA 99164, USA

* Correspondence: whfinch@bsu.edu

**Abstract:** Differential item functioning (DIF) is a critical step in providing evidence to support a scoring inference in building a validity argument for a psychological or educational assessment. Effect sizes can assist in understanding the accumulation of DIF at the test score level. The current simulation study investigated the performance of several proposed effect size measures under a variety of conditions. Conditions under study included varied sample sizes, DIF effect sizes, the proportion of items with DIF, and the type of DIF (additive vs. non-additive). DIF effect sizes under study included sDTF%, uDTF%, $\hat{\tau}_w^2$, $d$, $\overline{R}_\Delta^2$, $I_{DIF}^{2*}$, and $S - DIF - V$. The results of this study suggest that across study conditions, $\hat{\tau}_w^2$, $I_{DIF}^{2*}$, and $d$ were consistently the most accurate measures of the DIF effects. The effect sizes were also estimated in an empirical example. Recommendations and implications for practice are discussed.

**Keywords:** differential item functioning; effect sizes; measurement invariance; validity

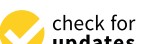



## 1. Introduction

A key requirement of educational measurement is that assessments be fair for individuals in different groups within the broader population [1]. Establishing test fairness involves the investigation of differential item/bundle functioning (DIF/DBF) present in the assessment. DIF refers to differential performance on assessment items for individuals in different groups who have the same level of the latent trait being measured, whereas DBF refers to differential group performance on sets, or bundles of items. When differential functioning is considered for the entire set of test items, we refer to differential test functioning (DTF). For the remainder of this paper, we will use DTF to refer to the presence of DIF in an entire assessment. DTF has been investigated in different contexts, from personality selection [2] to health care [3], including comparisons of effect sizes [2]. Capturing a lack of item invariance at the test level can assist in assessing if such issues influence score-level decisions and assisting with understanding the practical importance of a lack of invariance [4,5]. Given the centrality of assessment in educational systems, paramount importance must be given to ensuring fair and equal measurement of abilities for all individuals. DIF/DTF does not guarantee fairness or equal measurement or use of scores, but it does assist in providing a piece of evidence to support a path in that direction.

Test fairness and DIF has focused on item-level differences. However, given decisions about individuals are based on test scores, understanding the influence of DIF on overall test scores is critical [6]. Effect sizes at the score level can aid this process. A number of such effect sizes have been suggested, including the DIF variance estimate from a random effects model, both unweighted ($\hat{\tau}^2$) and weighted ($\hat{\tau}_w^2$) [7], Cohen's $d$ for between group average DIF [8,9], the average variance associated with DIF from logistic regression $\overline{R}_\Delta^2$; [8,10], the signed (sDTF) and unsigned (uDTF) DTF, and the percent scoring difference of the test (sDTF%/uDTF%). Doebler [11] described a new DIF assessment framework designed

to obviate the need to account for differences in group means on the measured latent trait (impact) or item purification. The statistics associated with this approach provide information about DTF, including a measure of the variance in differences between group difficulty parameters ($I_{DIF}^{2*}$), and a standardized DIF variance measure ($S-DIF-V$) for the difference between group difficulty parameters. These have not been studied for their effectiveness.

The remainder of this manuscript is organized as follows. First, a review of various methods for test-level DIF effect sizes is described, after which the goals of this study are presented. The simulation study used to assess the performance of these effect sizes is then described, followed by a presentation of the results for both a simulation and an applied data example. Finally, these results are discussed, and their implications for practice are considered. The simulation study was motivated by the ability to compare all effect sizes under known and controlled conditions, as has been common in other DTF effect size comparisons [2,12], and moves beyond comparisons that have only been on applied datasets [4].

### 1.1. Effect Size Methods for Quantifying the Magnitude of DIF in an Assessment

Effect size use in DIF detection has been emphasized for DIF identification [6] and for general measurement invariance work [13]. A number of effect size measures for quantifying DIF for a single assessment have been proposed. One such approach for quantifying the total amount of DIF in a set of items (e.g., subtests) is based on the concept of the random effect item response theory (IRT) model [7]. Consider the 2-parameter logistic model for DIF (2PL-DIF):

$$P(\theta) = \frac{exp\left(1.7a_i\left(\theta_j - b_i\right) + G\xi_i\right)}{1 + exp\left(1.7a_i\left(\theta_j - b_i\right) + G\xi_i\right)} \tag{1}$$

In Equation (1), $a_i$ is the discrimination parameter for item I, $b_i$ is the difficulty parameter for item i, $\theta_j$ is the latent trait value for examinee $j$, $G$ is group membership (focal group = 0 and reference group = 1), and $\xi_i$ = DIF effect for item $i$. In this context, $\xi_i = 0$ signals no DIF, $\xi_i > 0$ signals DIF against the focal group, and $\xi_i < 0$ signals DIF against the reference group. The log odds ratio from the Mantel–Haenszel test is an effective and appropriate estimate of $\xi_i$ [7] and can be modeled in the random effects context as follows:

$$log(\hat{\alpha}_{MHi}) = \mu + \xi_i + \varepsilon_i \tag{2}$$

where $\mu$ is the mean DIF across all items and $\varepsilon_i$ is the estimation error.

When no DIF is present, the variance of $\xi$, $\tau^2$, will be 0. In short, $\tau^2$ provides an estimate of the amount of DIF in a set of items, with larger values indicating a greater amount of cumulative DIF. Camilli and Penfield proposed two approaches to estimating $\tau^2$. The unweighted estimator takes the form:

$$\hat{\tau}^2 = \frac{\sum_{i=1}^{I}[log(\hat{\alpha}_{MHi}) - \hat{\mu}]^2 - \sum_{i=1}^{I} S_i^2}{I} \tag{3}$$

where $\hat{\mu}$ is the mean of $log(\hat{\alpha}_{MHi})$ across the $I$ items of the test, and $S_i^2$ is the variance in $log(\hat{\alpha}_{MHi})$ for each of the $i$ items of the test. The weighted estimator is as follows:

$$\hat{\tau}_w^2 = \frac{\sum_{i=1}^{I} w_i^2[log(\hat{\alpha}_{MHi}) - \hat{\mu}]^2 - \sum_{i=1}^{I} w_i^2}{\sum_{i=1}^{I} w_i^2} \tag{4}$$

where $w_i^2 = S_i^{-2}$. The index $\hat{\tau}_w^2$ can provide a more accurate estimation of the true amount of cumulative DIF (DTF) present in a set of items [7].

A second proposed method for quantifying the amount of DIF in an assessment involves a conversion of $log(\hat{\alpha}_{MHi})$ for item $i$ to a Cohen's $d$ effect size using a transformation

first proposed by Hasselblad and Hedges [9]. Specifically, assuming a logistic distribution and homogeneous variances, a log odds ratio, such as $log(\hat{\alpha}_{MHi})$ can be converted to $d$ by the following:

$$d = log(\hat{\alpha}_{MHi}) \frac{\sqrt{3}}{\pi} \tag{5}$$

In the context of comparing the amount of DTF present in two or more scales, we propose to calculate $log(\hat{\alpha}_{MHi})$ for each item in the test or item bundle, convert these to $d$ using Equation (5), and then average the $d$s across the items, creating $\bar{d}$. The potential advantage is that $d$ is on a familiar scale to researchers and has interpretive values for application. Two values of $\bar{d}$ will be calculated in this study, $\bar{d}_s$ in which the signs of the $d$ values are taken into consideration, and $\bar{d}_u$, where the absolute value of the $d$s is used.

A third approach is based on logistic regression (LR), a popular method for DIF detection [14]. When LR is used for DIF assessment, $R^2_\Delta$, the effect size, is reported with the statistical test for each item. $R^2_\Delta$ reflects the amount of DIF present for the item. To characterize the magnitude of uniform DIF for an item, this statistic is calculated as the difference between the $R^2$ associated with the model including only the latent trait as the predictor for the item response, and the $R^2$ associated with the model including the latent trait and an indicator for the grouping variable. In order to estimate the amount of DIF present for an entire scale, we propose the use of the mean of $R^2_\Delta$, $\overline{R}^2_\Delta$, taken across items in an item bundle or an entire assessment. Thus, for the set of items of interest, $R^2_\Delta$ will be calculated for each item and $\overline{R}^2_\Delta$ will be taken across the set of items, with larger values reflecting a greater amount of DIF present in the set of items. Note that this estimate of DTF is unsigned and does not reflect cancellation DIF.

Doebler [11] described DIF effect sizes that have shown promise for assessing the magnitude of DIF at the item level, and that may be useful for describing the impact of DIF at the test level. One of these statistics is based upon the variance in the difference of item level parameter estimates (e.g., item difficulty) across groups. For the two groups' case, this difference is estimated as follows:

$$\hat{\Delta}\beta_i = \hat{\beta}_{Fi} - \hat{\beta}_{Ri} \tag{6}$$

where

$\hat{\beta}_{Fi}$ = Focal group difficulty estimate for item i
$\hat{\beta}_{Ri}$ = Reference group difficulty estimate for item i

When uniform DIF is not present for an item, then $\hat{\Delta}\beta_i = 0$. The variance associated with this statistic can be used to estimate the total amount of DIF present in a test. One such example is the $I^2_{DIF}$ statistic, which is based upon a statistic used in meta-analysis and which can be calculated as follows:

$$I^2_{DIF} = \frac{\hat{v}^2}{\hat{v}^2 + s^2_{\Delta\beta}} \tag{7}$$

where

$\hat{v}^2$ = Variance in the $\hat{\Delta}\beta_i$
$s^2_{\Delta\beta}$ = Mean squared error of $\hat{\Delta}\beta_i$
The $s^2_{\Delta\beta}$ value is estimated as

$$s^2_{\Delta\beta} = \frac{(I-1)\sum_{i=1}^{I} w_i}{\sum_{i=1}^{I} w_i{}^2 - \sum_{i=1}^{I} w_i^2} \tag{8}$$

where

$I$ = Number of items
$w_i = \frac{1}{s^2_{\hat{\Delta}\beta_i}}$

Larger values of $I^2_{DIF}$ indicate greater inconsistency in the DIF structure across items.

In practice, the REML estimator of $\hat{v}^2$ using maximum likelihood cannot be obtained in closed form. An alternative estimator of $I^2_{DIF}$ can be calculated using an estimate of $\hat{v}^2$ proposed by DerSimonian and Laird (1986). This estimator takes the form:

$$I^{2*}_{DIF} = \frac{max(0, Q - (I - 1))}{Q} \tag{9}$$

where

$$Q = \sum_{i=1}^{I} \frac{\left(\hat{\Delta}\beta_i - \Delta\right)^2}{s^2_{\Delta\beta_i}}$$

$$\Delta = \frac{\left(\sum_{i=1}^{I} w_i \hat{\Delta}\beta_i\right)}{\sum_{i=1}^{I} w_i}$$

$I^{2*}_{DIF}$ can be interpreted as the ratio of excess homogeneity to the total heterogeneity in DIF across the items, and serves as an estimate of $I^2_{DIF}$ such that larger values indicate a higher level of DIF in the assessment being studied.

Chalmers et al. [15] described two approaches for characterizing the amount of DIF in an assessment, both based on the group-specific expected test score function in Equation (10).

$$T(\theta, \psi_G) = \sum_{j=1}^{n} S_j(\theta, \psi_G) \tag{10}$$

where

$\psi_G$ = Item parameters for group G
$S_i = \sum_{K=0}^{K-1} k \cdot P(y = k|\theta, \psi_j)$; score function for item i
$y$ = Item response
$k$ = Item category
$\theta$ = Latent trait value

In turn, $T(\theta, \psi_G)$ can be used to ascertain whether there is differential test functioning (DTF) across multiple groups. One such statistic is an index of signed DTF.

$$sDTF = \int [T(\theta, \psi_R) - T(\theta, \psi_F))]g(\theta)d(\theta) \tag{11}$$

where

$g(\theta)$ = Weight function such that $\int g(\theta)d(\theta) = 1$
$\psi_R$ = Reference group item parameters
$\psi_F$ = Focal group item parameters

The sample estimate of $sDTF$ is obtained by evaluating the function in Equation (11) at a set of $Q$ quadrature points, as in Equation (12).

$$sDTF = \sum_{q=1}^{Q} T(X_q, \psi_R) - T(X_q, \psi_F))g(X_q) \tag{12}$$

where

$X_q$ = Quadrature value
$g(X_q)$ = Weight associated with the quadrature node

Researchers can interpret $sDTF$ as the mean difference between the test response functions for the two groups, with negative values indicating that the reference group had lower mean scores on the assessment and positive values indicating that the focal group had lower mean scores, after accounting for item parameter values.

An alternative to the $sDTF$ is an unsigned measure of DTF, which focuses on group differences in the test function, without respect to which group is favored. This unsigned DTF measure is expressed as follows:

$$uDTF = \int |T(\theta, \psi_R - T(\theta, \psi_F))|g(\theta)d(\theta) \tag{13}$$

Note that the *uDTF* is identical to the *sDTF* with the exception that the absolute value of test function differences is applied in Equation (13). As with the *sDTF*, quadrature is used to obtain estimates for specific samples, as in Equation (14):

$$uDTF = \sum_{q=1}^{Q} |T(X_q, \psi_R) - T(X_q, \psi_F))| g(X_q) \tag{14}$$

*uDTF* reflects the mean difference between the two groups' curves, and can be standardized in order to make interpretation easier, as in Equation (15).

$$uDTF\% = \frac{uDTF}{TS} \times 100 \tag{15}$$

where

$TS$ = Highest possible test score

Larger values of these statistics reflect a greater degree of DTF.

Doebler [11] described another effect size measure of DIF across a set of items, the standardized DIF variance ($S - DIF - V$). This statistic takes the form:

$$S - DIF - V = \frac{(J_R + J_F - 2)\hat{\tau}^2}{(J_F - 1)\sigma_F^2 + (J_F - 1)\sigma_F^2} \tag{16}$$

where

$J_R$ = Reference group sample size

$J_F$ = Focal group sample size

$\sigma_F^2$ = Variance in latent trait for focal group

$\sigma_R^2$ = Variance in latent trait for reference group

Doebler argues that in contrast to $I_{DIF}^{2*}$, $S - DIF - V$ is largely independent of sample size and thus may be particularly useful as an effect size.

There is a dearth of work examining the performance of these test-level DIF (DTF) effect size measures. Finch et al. [8] found that when comparing the amount of DIF present in two assessments of the same construct $\hat{\tau}^2$, $\bar{d}_s$, $\bar{d}_u$, and $\overline{R}_\Delta^2$ all were effective in identifying the test with greater DIF across its items. In contrast, when the tests were not of equal length, $\hat{\tau}^2$ yielded more accurate results compared to other methods studied. Furthermore, when DIF was present for both tests but greater in one than the other, $\hat{\tau}^2$ was better able to identify which had greater levels of DIF than were $\bar{d}_s$, $\bar{d}_u$, and $\overline{R}_\Delta^2$. With respect to $I_{DIF}^{2*}$ and $S - DIF - V$, Doebler [11] demonstrated their use with an extant dataset and supported that they were able to identify cumulative DIF across a set of items in a scale.

### 1.2. Study Goals

This study's goals were to (a) describe a variety of effect size statistics designed to measure the impact of uniform DIF at the item level in terms of the impact at the score level, (b) compare these measures across simulated conditions, and (c) show these effect sizes in an applied example. A simulation was used to compare the effect sizes following past research [2,5]. In this work, population parameter effect size values were obtained by generating populations consisting of 20,000,000 individuals in each group under a specific DIF condition. The population data generating conditions appear in Table 1 and are described in the methods section below. This study adds to the literature by comparing a number of test-level DIF effect sizes, several of which have not been examined heretofore using a simulation methodology.

**Table 1.** Study factors and conditions used in the simulation.

| Factor | Levels |
| --- | --- |
| Number of items | 10, 20 |

**Table 1.** *Cont.*

| Factor | Levels |
|---|---|
| Group sample sizes | 250/250, 500/500, 1000/1000, 2000/2000, 500/250, 1000/250, 2000/250, 1000/500, 2000/500, 2000/1000 |
| Group mean difference | 0, −0.5, −1 |
| Proportion of DIF items | 0, 0.1, 0.2 |
| Uniform DIF magnitude | 0, 0.4, 0.8 |
| DIF effect | Additive, Non-additive |
| Effect sizes | sDTF%, uDTF%, $\hat{\tau}_w^2$, $d$, $\overline{R}_\Delta^2$ from logistic regression, $I_{DIF}^{2*}$, $S-DIF-V$ |

## 2. Materials and Methods

A Monte Carlo simulation study (1000 replications per condition) was conducted to compare the effect size measures with one another across a variety of study conditions. Item response data for dichotomous items were generated using a Rasch model in order to follow the modeling used in Doebler [11]. The study conditions, which are described below and appear in Table 1, were completely crossed, and were selected to represent what is common in many DIF studies.

Data generation and analysis were completed with the R software system, version 4.0 [16]. The item difficulty parameters used to generate data for the reference group were drawn from a widely used intelligence test [17] and appear in Table 2. The method used to induce DIF for the focal group is described below.

**Table 2.** Data generating item difficulty parameter values for the two simulated assessments in the 30-item condition.

| Item | Reference Group Item Difficulty Parameter Values |
|---|---|
| 1 | −1.92 |
| 2 | −0.96 |
| 3 | −0.55 |
| 4 | −0.14 |
| 5 | 0.01 |
| 6 | 0.22 |
| 7 | 0.41 |
| 8 | 0.59 |
| 9 | −0.60 |
| 10 | 0.23 |
| 11 | 0.77 |
| 12 | 0.81 |
| 13 | 0.92 |
| 14 | 0.93 |
| 15 | 1.02 |
| 16 | 1.15 |
| 17 | 1.30 |
| 18 | 1.37 |
| 19 | 1.68 |
| 20 | 1.74 |

*2.1. Manipulated Factors*

2.1.1. Number of Items

Prior research has shown that the assessment of DIF/DTF can be influenced by the test length, with longer tests leaving potentially more items for better matching of groups for DIF detection [18–21]. For this reason, two test lengths were simulated in this study (10 and 20 items), representing short- to moderate-length tests. These values are common

in practice, especially with psychological scales [22–24], and have appeared in previous simulation studies within this range examining DIF detection [18–20,25].

### 2.1.2. Percentage, Magnitude, and Type of DIF

The percentage of DIF occurred at three levels, 0%, 10%, and 20%, found in practice [26]. DIF was simulated by adjusting the item difficulty parameter values for the focal group through manipulation of the area between the item response functions via Raju's formula. These differences were employed in order to simulate no DIF (0), small (0.4), and large (0.8) DIF [21]. The percent of 0% DIF condition was by necessity not crossed with the level of DIF conditions. As noted above, the item difficulty values used to generate the data for the reference group appear in Table 2. When DIF was present, the magnitude values listed above were added to the values for items 2, 3, 6, and 7 (for the appropriate proportion of items with DIF) in Table 2 for the target items in order to obtain item difficulty parameters for the focal group. These items were selected randomly. DIF was simulated to be either additive (all DIF items favored the same group) or non-additive (half of the DIF items favored the reference group, and half favored the focal group).

### 2.1.3. Sample Size/Sample Size Ratio

Focal and reference group sample sizes were simulated at a variety of levels, as seen in Table 1. These values represent both the applied and simulation literature on DIF [14,21,27,28].

### 2.1.4. Group Ability Differences (Impact)

Group mean differences in the latent trait are associated with Type I error inflation for uniform DIF [27,29–31]. Therefore, to ascertain the influence of impact on the performance of the effect sizes, mean abilities of the focal and reference groups were set to be 0/0, $-0.5/0$, or $-1/0$.

### 2.1.5. Study Outcomes

The study outcomes of interest were the mean effect size values, the empirical standard errors for the effect sizes, and the relative estimation bias of the effect size measures. The empirical standard error is the standard deviation of the 1000 parameter estimates for each combination of conditions. In other words, for each combination of the study conditions, the standard deviation of each effect size measure was calculated for the 1000 estimated values. Likewise, the mean effect size value for each effect size statistic was calculated across the 1000 parameter estimates for each combination of conditions. The relative bias was calculated as follows:

$$RB = \frac{\hat{\theta} - \theta}{\theta} \tag{17}$$

where

$\theta$ = Population parameter value
$\hat{\theta}$ = Parameter estimate

The population parameter effect size values were obtained by generating population simulations consisting of 20,000,000 individuals in each group under a specific DIF condition (e.g., 0.8 magnitude with 20% of items containing DIF). To ensure the stability of these values, this procedure was conducted 3 times for each condition, and it was noted that the resulting effect size values were virtually identical across these replications. In order to assess which main effects and interactions of the conditions were related to the relative bias, analysis of variance (ANOVA), in conjunction with the $\omega^2$, was used, where values above 0.10 singled out factors to examine.

With respect to interpretation of the outcomes, the RB is most useful for comparison across the various effect size statistics because it accounts for differences in the scale of each statistic by standardizing it to the population value. In other words, for each effect size measure, RB reflects the same metric, difference between the estimated and population

values divided by the population value. The empirical standard error is most useful for comparisons within the same effect size statistic across different simulation conditions. It is not helpful for comparing across effect size measures because they are calculated on different scales. Finally, the mean of the raw effect size statistic is also useful for comparison across simulation conditions within effect size measure and not across effect sizes, again due to differences in scale.

## 3. Results

### 3.1. No DIF

When there was no DIF present in the population, there were no statistically significant terms associated with the values of effect sizes. Table 3 contains the means and empirical standard errors of the DTF effect size statistics in the absence of the DIF condition. If DIF was not present, values of 0 should be observed for these effect sizes. None of these statistics yielded values that were exactly 0, with all but the sDTF% exhibiting a positive bias. Note that it is not possible to calculate relative bias in the no DIF condition because it would require division by 0.

**Table 3.** DTF effect size statistics and standard error when DIF was absent.

| Statistic | Mean | Standard Error |
|---|---|---|
| $I^{2*}_{DIF}$ | 0.04 | 0.11 |
| $S-DIF-V$ | 0.08 | 0.04 |
| $\hat{\tau}^2_w$ | 0.003 | 0.02 |
| $\overline{R}^2_\Delta$ | 0.001 | 0.003 |
| $d$ | 0.01 | 0.02 |
| uDTF% | 2.08 | 2.88 |
| sDTF% | −0.48 | 3.23 |

### 3.2. DIF Present

When DIF was simulated to be present among the items, ANOVA results revealed that the interactions of the proportion of DIF items by DIF magnitude ($F_{1,441} = 48.77$, p < 0.0001, $\omega^2 = 0.094$), impact by type of DIF ($F_{2,441} = 12.07$, $p < 0.0001$, $\omega^2 = 0.073$), and the interaction magnitude of DIF by type of DIF ($F_{1,441} = 5.74$, $p = 0.017$, $\omega^2 = 0.013$) were associated with relative bias. Table 4 includes the relative bias, effect size means, and standard errors across simulation replications for the proportion of DIF items by the magnitude of DIF. Relative biases of $\hat{\tau}^2_w$, $d$, and sDTF% were generally uninfluenced by the magnitude of DIF or the proportion of items that contained DIF. In contrast, $I^{2*}_{DIF}$, $S-DIF-V$, and $\overline{R}^2_\Delta$ yielded different levels of relative bias based on DIF magnitude and proportion. More specifically, $I^{2*}_{DIF}$ yielded somewhat higher levels of bias for the lower DIF magnitude, and $S-DIF-V$ had a higher bias for larger DIF magnitudes. The $\overline{R}^2_\Delta$ effect size yielded higher levels of relative bias for both greater DIF magnitude and a larger proportion of DIF items. In addition, for each of the effect sizes, mean values were larger when a greater proportion of items exhibited DIF and when the magnitude of DIF was greater.

The empirical standard errors by DIF magnitude and proportion of DIF items, as well as for the No DIF condition, appear in the third panel of Table 4.

For $S-DIF-V$ and $\overline{R}^2_\Delta$ the standard errors in the No DIF case were lower than when DIF was present. In contrast, $I^{2*}_{DIF}$, uDTF% and sDTF% exhibited empirical standard errors that were equal to or greater than the empirical standard errors when DIF was present. The standard errors for $\hat{\tau}^2_w$ and $d$ were not impacted by the presence of DIF. When DIF was present, the empirical standard errors for $I^{2*}_{DIF}$, uDTF%, and sDTF% were larger for a higher proportion of DIF items when the magnitude of DIF was 0.4, whereas, for a magnitude of 0.8, the reverse pattern was in evidence. The results for $S-DIF-V$ were the opposite of these three variables, and for $\hat{\tau}^2_w$, $d$, and $\overline{R}^2_\Delta$ there appeared to be little to

no relationship between either the DIF magnitude or proportion of DIF items and the empirical standard error.

**Table 4.** Effect size values, empirical standard error, and relative estimation bias by magnitude of DIF and proportion of items with DIF.

| Mag DIF | Prop DIF | $I^{2*}_{DIF}$ | $S-DIF-V$ | $\hat{\tau}^2_w$ | $\overline{R}^2_\Delta$ | $d$ | uDTF% | sDTF% |
|---|---|---|---|---|---|---|---|---|
| 0.4 | 0.1 | 0.57 | 0.12 | 0.02 | 0.004 | 0.10 | 13.40 | −13.38 |
|  | 0.2 | 0.64 | 0.19 | 0.06 | 0.008 | 0.12 | 21.64 | −17.59 |
| 0.8 | 0.1 | 0.61 | 0.22 | 0.05 | 0.011 | 0.11 | 22.04 | −16.80 |
|  | 0.2 | 0.70 | 0.25 | 0.08 | 0.012 | 0.14 | 25.12 | −18.85 |
| *Empirical standard error* | | | | | | | | |
| 0.4 | 0.1 | 0.08 | 0.06 | 0.02 | 0.01 | 0.02 | 2.18 | 2.94 |
|  | 0.2 | 0.11 | 0.05 | 0.01 | 0.01 | 0.02 | 2.78 | 3.20 |
| 0.8 | 0.1 | 0.09 | 0.06 | 0.02 | 0.01 | 0.02 | 2.66 | 3.17 |
|  | 0.2 | 0.07 | 0.07 | 0.02 | 0.01 | 0.02 | 2.60 | 3.15 |
| *Relative estimation bias* | | | | | | | | |
| 0.4 | 0.1 | 0.02 | 0.01 | 0.003 | −0.06 | −0.01 | −0.02 | −0.01 |
|  | 0.2 | 0.02 | 0.02 | 0.003 | −0.08 | −0.01 | −0.01 | −0.01 |
| 0.8 | 0.1 | 0.01 | 0.04 | 0.003 | −0.07 | −0.01 | −0.01 | −0.01 |
|  | 0.2 | 0.01 | 0.04 | 0.002 | −0.08 | −0.01 | −0.02 | −0.01 |

Table 5 includes the relative bias, mean effect size values, and standard errors by DIF magnitude and type of DIF.

**Table 5.** Effect size values, empirical standard error, and relative estimation bias by magnitude of DIF and type of DIF.

| Mag DIF | DIF Type | $I^{2*}_{DIF}$ | $S-DIF-V$ | $\hat{\tau}^2_w$ | $\overline{R}^2_\Delta$ | $d$ | uDTF% | sDTF% |
|---|---|---|---|---|---|---|---|---|
| 0.4 | Add | 0.63 | 0.18 | 0.04 | 0.007 | 0.12 | 18.80 | −13.39 |
|  | Non | 0.58 | 0.13 | 0.04 | 0.005 | 0.11 | 17.59 | −7.84 |
| 0.8 | Add | 0.73 | 0.35 | 0.09 | 0.02 | 0.14 | 22.98 | −17.10 |
|  | Non | 0.57 | 0.14 | 0.09 | 0.02 | 0.11 | 21.41 | −8.45 |
| *Empirical standard error* | | | | | | | | |
| 0.4 | Add | 0.10 | 0.05 | 0.02 | 0.005 | 0.02 | 2.56 | 3.21 |
|  | Non | 0.09 | 0.06 | 0.02 | 0.006 | 0.02 | 2.47 | 2.91 |
| 0.8 | Add | 0.09 | 0.05 | 0.02 | 0.006 | 0.02 | 2.76 | 3.34 |
|  | Non | 0.05 | 0.09 | 0.02 | 0.009 | 0.02 | 2.46 | 2.91 |
| *Relative estimation bias* | | | | | | | | |
| 0.4 | Add | 0.03 | 0.02 | 0.003 | −0.07 | −0.02 | −0.02 | −0.01 |
|  | Non | 0.01 | 0.01 | 0.003 | −0.07 | −0.01 | −0.01 | −0.01 |
| 0.8 | Add | 0.02 | 0.06 | 0.003 | −0.08 | −0.01 | −0.02 | −0.01 |
|  | Non | 0.01 | 0.02 | 0.003 | −0.07 | −0.01 | −0.01 | −0.01 |

When DIF was additive in nature (i.e., all DIF items favored the same group), the relative bias was consistently higher for $I^{2*}_{DIF}$, $S-DIF-V$, and uDTF%. This effect was particularly marked for $S-DIF-V$ in the higher DIF condition. In contrast, relative bias in $\hat{\tau}^2_w$, $d$, and sDTF% were not affected by either DIF magnitude or type. Finally, $\overline{R}^2_\Delta$ yielded the highest relative bias across conditions.

With respect to the mean estimates, each effect size yielded larger values when the magnitude of DIF was larger. This result was demonstrated in both the mean effect size values as well as the ratios of these values to the no DIF case. When DIF was not additive (i.e., equal numbers of items favored the two groups), $I^{2*}_{DIF}$, $S-DIF-V$, and $d$ displayed little to no differences in the mean values between the two DIF magnitude levels. In addition, the difference in mean values between DIF magnitudes of 0.4 and 0.8 was less for uDTF% and sDTF% for non-additive DIF. Finally, $\hat{\tau}^2_w$ and $\overline{R}^2_\Delta$ both appeared to be largely unaffected by DIF characteristics, such that the relationship of DIF magnitude on the mean effect size value was similar for both types of DIF.

The empirical standard errors for the effect sizes by DIF magnitude and type also appear in Table 5. The overall patterns of relative empirical standard error sizes for the No DIF and DIF conditions by effect size statistic that were discussed with respect to Table 4 are also evident in Table 5 as well. In terms of the combination of DIF magnitude and type, there was no impact on the empirical standard errors of either $\hat{\tau}^2_w$ or $d$. The standard errors for $S-DIF-V$ and $\overline{R}^2_\Delta$ were larger for the non-additive DIF condition, and this pattern was stronger when the DIF magnitude was 0.8. In contrast, the empirical standard errors for $I^{2*}_{DIF}$, uDTF% and sDTF% were smaller in the non-additive DIF case, and this difference with the additive DIF condition was more pronounced for a greater DIF magnitude.

Table 6 includes the relative estimation bias, mean effect size values, and empirical standard errors by impact and type of DIF.

**Table 6.** Effect size values, empirical standard error, and relative estimation bias by impact and type of DIF.

| Impact | DIF Type | $I^{2*}_{DIF}$ | $S-DIF-V$ | $\hat{\tau}^2_w$ | $\overline{R}^2_\Delta$ | $d$ | uDTF% | sDTF% |
|---|---|---|---|---|---|---|---|---|
| 0 | Add | 0.66 | 0.24 | 0.06 | 0.01 | 0.13 | 18.34 | −12.80 |
| | Non | 0.61 | 0.17 | 0.06 | 0.01 | 0.12 | 15.30 | −5.44 |
| −0.5 | Add | 0.67 | 0.25 | 0.06 | 0.01 | 0.12 | 20.73 | −17.07 |
| | Non | 0.59 | 0.14 | 0.05 | 0.01 | 0.12 | 19.01 | −11.29 |
| −1.0 | Add | 0.71 | 0.29 | 0.06 | 0.01 | 0.13 | 24.19 | −20.68 |
| | Non | 0.53 | 0.10 | 0.05 | 0.01 | 0.10 | 23.59 | −17.89 |
| | | | | Standard error | | | | |
| 0 | Add | 0.09 | 0.06 | 0.02 | 0.005 | 0.02 | 2.45 | 3.38 |
| | Non | 0.08 | 0.07 | 0.02 | 0.007 | 0.02 | 2.41 | 2.90 |
| −0.5 | Add | 0.09 | 0.05 | 0.02 | 0.006 | 0.02 | 2.71 | 3.21 |
| | Non | 0.07 | 0.07 | 0.02 | 0.007 | 0.02 | 2.47 | 2.88 |
| −1.0 | Add | 0.11 | 0.04 | 0.02 | 0.005 | 0.02 | 2.86 | 3.26 |
| | Non | 0.07 | 0.08 | 0.02 | 0.007 | 0.02 | 2.56 | 2.96 |
| | | | | Relative estimation bias | | | | |
| 0 | Add | 0.01 | 0.02 | 0.01 | −0.06 | −0.01 | −0.01 | −0.01 |
| | Non | 0.01 | 0.03 | 0.01 | −0.07 | −0.01 | −0.01 | −0.01 |
| −0.5 | Add | 0.03 | 0.05 | 0.01 | −0.07 | −0.02 | −0.02 | −0.01 |
| | Non | −0.01 | 0.01 | 0.01 | −0.08 | −0.01 | −0.01 | −0.01 |
| −1.0 | Add | 0.04 | 0.07 | 0.01 | −0.08 | −0.01 | −0.03 | −0.01 |
| | Non | −0.02 | −0.02 | 0.01 | −0.06 | −0.02 | −0.01 | −0.01 |

As was evident in Table 5, the relative biases of $\hat{\tau}^2_w$, d, and sDTF% were affected relatively little by the type of DIF, regardless of the level of impact. In contrast, the type of DIF did influence the values that were obtained for the other statistics to at least some extent. More specifically, the type of DIF had a greater influence on the relative bias with greater levels of impact for $I^{2*}_{DIF}$, $S-DIF-V$ and uDTF%. Across conditions, $\overline{R}^2_\Delta$ yielded the highest degree of relative bias from among the effect sizes included in this study.

A similar overall pattern of results was evident for the mean values of the effect sizes themselves. For example, $\overline{R}_{\Delta}^2$ was unaffected by the level of impact. Furthermore, $\hat{\tau}_w^2$ and $d$ evinced relatively small differences in mean values across the different levels of impact. In contrast, magnitudes for several of the effect size measures were impacted differentially by the combination of impact and DIF type. As an example, the values of $I_{DIF}^{2*}$ and $S-DIF-V$ increased concomitantly with greater levels of impact when the type of DIF was additive. Conversely, when the type of DIF was non-additive, the values of these two effect sizes decreased with increasing levels of impact. In other words, both $I_{DIF}^{2*}$ and $S-DIF-V$ were less likely to reflect the presence of DIF when there was a larger impact and when the type of DIF was non-additive, but were more likely to reflect the presence of more DIF among the items when the type of DIF was additive, and the level of impact was higher. In contrast to this pattern, the results in Table 6 show that for both uDTF% and sDTF%, there was an increase in values as the level of impact increased as well. However, this increase was greater for the non-additive DIF, as reflected in both the means and ratios to the non-DIF condition.

The empirical standard errors for the effect size statistics by impact and type of DIF appear in the third panel of Table 6, along with the standard errors when no DIF was present. As was true in the other tables, there was no relationship between either impact or the type of DIF and the empirical standard errors for $\hat{\tau}_w^2$ or $d$. In addition, the standard errors for $\overline{R}_{\Delta}^2$ were larger for the non-additive DIF condition, and this result was consistent across levels of impact. As was the case in Table 4, the empirical standard errors for the additive DIF condition were larger for both uDTF% and sDTF%, and this difference was larger for uDTF% when greater amounts of impact were present. Effect size values, empirical standard errors, and relative estimation bias by sample size (Supplemental Table S1) and effect size values, empirical standard errors, and relative estimation bias by number of items (Supplemental Table S2) appear in Supplemental Materials.

### 3.3. Empirical Example

In order to demonstrate the use of these effect size statistics in an applied setting, DIF analysis was conducted for an empirical example, following past examples in this area [2]. The data used for this purpose were drawn from the verbal aggression dataset that is included in the difR package [32]. The data include 316 responses to a 24-item verbal aggression questionnaire. The items asked respondents about four different scenarios (bus fails to stop for me, miss train because clerk provided faulty information, grocery store closes as I am about to enter, and operator disconnects me when I used my last 10 cents for a call). For each scenario, the respondents are also asked whether they would curse, scold, or shout or would want to (but not act on a) curse, scold, or shout. The combination of these scenarios (4) and potential actions (6) yields 24 items. For each item, individuals respond either yes (1) or no (0). Uniform DIF for these items was assessed by both the Mantel–Haenszel test and logistic regression based on gender, with 243 women and 73 men in the sample. The effect sizes featured in the simulation study were applied to the verbal aggression dataset.

Table 7 includes the difficulty parameter estimates, ETS $\Delta$ effect size, ETS DIF classification, and the $\overline{R}_{\Delta}^2$ effect size from logistic regression for each item in the verbal aggression scale. Statistically significant DIF is denoted by *.

Based on these results, items 6, 12, 16, 17, 19, and 20 were identified with statistically significant uniform DIF by MH. The LR results identified items 4, 6, 14, 16, 17, and 19 as having gender-based DIF. The difficulty parameter estimates were lower for females on items 4, 6, and 12, whereas they were higher for females on items 16, 17, 19, and 20. Higher item difficulty means that an individual needs a higher degree of the latent trait (verbal aggression) to endorse the item.

The DTF effect size estimates for the verbal aggression data appear in Table 8.

**Table 7.** Item difficulty estimates by group, and item level DIF results for verbal anger data.

| Item | Female Item Difficulty | Male Item Difficulty | ETS Δ * | MH α | Logistic Regression $\overline{R}^2_\Delta$ |
|------|------------------------|----------------------|---------|------|---------------------------------------------|
| 1 | −1.26 | −1.11 | −1.25 | 1.70 | 0.007 |
| 2 | −0.59 | −0.49 | −1.34 | 1.77 | 0.006 |
| 3 | −0.11 | 0.02 | −0.87 | 1.45 | 0.007 |
| 4 | −1.85 | −1.46 | −1.56 | 1.94 | 0.020 * |
| 5 | −0.76 | −0.56 | −1.61 | 1.98 | 0.016 |
| 6 | −0.18 | 0.53 | −2.49 * | 2.88 | 0.034 * |
| 7 | −0.50 | −0.64 | 0.14 | 0.94 | 0.000 |
| 8 | 0.86 | 0.16 | 0.77 | 0.72 | 0.005 |
| 9 | 1.47 | 1.74 | −1.00 | 1.53 | 0.008 |
| 10 | −1.13 | −0.95 | −1.23 | 1.69 | 0.007 |
| 11 | 0.43 | 0.09 | −0.20 | 1.09 | 0.000 |
| 12 | 0.96 | 1.34 | −2.00 * | 2.35 | 0.013 |
| 13 | −1.11 | −1.65 | 0.53 | 0.80 | 0.001 |
| 14 | −0.18 | −1.11 | 1.63 | 0.50 | 0.011 * |
| 15 | 0.89 | 0.83 | −0.38 | 1.18 | 0.002 |
| 16 | −0.62 | −1.85 | 2.67 * | 0.32 | 0.024 * |
| 17 | 0.34 | −0.87 | 2.31 * | 0.38 | 0.025 * |
| 18 | 1.59 | 1.17 | 0.55 | 0.79 | 0.000 |
| 19 | 0.48 | −0.64 | 1.82 * | 0.46 | 0.023 * |
| 20 | 1.78 | 0.77 | 1.76 * | 0.47 | 0.017 |
| 21 | 3.16 | 2.50 | 1.06 | 0.64 | 0.003 |
| 22 | −0.55 | −1.28 | 1.03 | 0.64 | 0.006 |
| 23 | 0.57 | −0.20 | 1.05 | 0.64 | 0.007 |
| 24 | 1.98 | 2.08 | −1.11 | 1.61 | 0.004 |

* Statistically significant ($\alpha$ = 0.05) uniform DIF.

**Table 8.** DTF effect size statistics for verbal anger data.

| Statistic | Full Verbal Aggression Data | Non-DIF Verbal Aggression Data |
|-----------|-----------------------------|--------------------------------|
| $I^{2*}_{DIF}$ | 0.78 | 0.54 |
| $S-DIF-V$ | 0.17 | 0.08 |
| $\hat{\tau}^2_w$ | 0.24 | 0.05 |
| $\overline{R}^2_\Delta$ | 0.01 | 0.004 |
| $d$ | 0.34 | 0.01 |
| uDTF% | 62.46 | 34.08 |
| sDTF% | −36.29 | −16.53 |

Effect size values were calculated for both the full set of 24 items and for the subset of 17 items that were not found to have statistically significant DIF. The purpose of this latter analysis was to provide a point of comparison for the two scenarios (with and without DIF). It is clear from these results that when DIF was present, the effect size values were larger than when no DIF was present. It should also be noted that these effect sizes are each designed such that the number of items should not be a factor with respect to their magnitude. Therefore, differences in effect size value reflect differences in the amount of collective DIF present in the items.

## 4. Discussion

Assessment fairness and equity is an increasingly important issue in educational and psychological measurement. In order for scores from such assessments to be meaningful at the individual and group levels, the measures themselves must perform in the same way for individuals from across the population. When this is not the case, and an instrument performs differentially for individuals from different subgroups, the scores cannot be assumed to be valid reflections of the construct being measured. Traditionally, DIF detection studies have focused primarily on identifying individual items that might exhibit DIF so that they can then be edited or removed. Relatively less attention has been paid to assessing the impact of DIF at the total scale level. Recently, however, authors have suggested such measures for use in ascertaining the extent of the impact of DIF on the total test score [8,11], as well as determining guidelines for the size of the effects and when invariance is of

practical importance [3]. The focus of this study was on comparing the performance of several such effect sizes in order to gain insights into which of them might provide measurement professionals the greatest insight into DIF impacts at the test level, and under what conditions.

The results presented above suggest that several of the effect sizes included in this study are potentially useful tools for researchers interested in assessing the effect of DIF across items on an instrument as a whole. In particular, it appears that $\hat{\tau}_w^2$, sDTF%, and $d$ were perhaps the best performers when considering the totality of the evidence. When DIF was not present, $\hat{\tau}_w^2$ exhibited the lowest degree of bias from the population value of 0, with $d$ also demonstrating relatively low bias. In addition, when DIF was present in the population, these three statistics had relatively low standard errors and low relative bias. In addition, these effect size statistics were largely impervious to group differences in the mean of the latent trait being measured by the assessment.

## 5. Directions for Future Research

Additional research needs to be conducted with an eye toward developing guidelines for interpreting the effect sizes studied here. Appropriate guidelines in the DIF literature, and in particular DTF, have been a challenge and will continue to be as new DIF procedures are proposed. In addition, the guidelines may differ by the type of outcome studied (e.g., observed vs. latent scores [5]). Future work should focus on a greater array of DIF magnitudes and proportions of items than were included in the current study. In addition, researchers may also want to consider developing confidence intervals for these effect sizes. In particular, a bootstrap approach to estimating standard errors that could then be used to estimate confidence intervals would be useful in providing researchers with greater information about the actual impact of DIF on the total assessment. Given the popularity of polytomous items in the social sciences, the performance of these statistics should be examined for items beyond the dichotomous case presented here, with appropriate modifications made where necessary. In this same vein, future research should also include a greater variety of dichotomous data-generating models, such as the 2-parameter logistic and 3-parameter logistic. The items selected to have DIF in the current study tended to be relatively easy, with the most difficult having a population difficulty parameter value of 0.41. Further simulations with more difficult items exhibiting DIF would be useful for future studies. Work focused on a different set of DIF magnitude values would also be quite useful. Although featured in previous research [21], the DIF magnitudes selected for this study were relatively large. Therefore, it would be useful for practitioners if future research examined the sensitivity of these effect size statistics to cases when DIF was smaller (e.g., 0.1 or 0.2).

## 6. Conclusions

The results presented above demonstrate that there exist multiple useful statistics for characterizing the effect of DIF at the full assessment level. These effect sizes should be useful tools for researchers as they consider the impact of DIF on a scale in order to determine whether, in fact, it is problematic. These results show that three of the statistics studied here, $\hat{\tau}_w^2$, $I_{DIF}^{2*}$, and $d$ may prove to be particularly useful. Therefore, measurement practitioners are encouraged to use them in practice, and psychometricians are encouraged to give them further study in order to develop guidelines for interpreting them. These effect sizes should serve as informative additions to DIF studies by providing information about the total impact of DIF on the total score taken from a scale. It is hoped that the current study has provided useful information in that regard, as well as a roadmap for future work in this area.

**Supplementary Materials:** The following supporting information can be downloaded at https://www.mdpi.com/article/10.3390/psych5010013/s1. Table S1, Effect size values by sample size when DIF was present; Table S2, Effect size values by number of items when DIF was present.

**Author Contributions:** Conceptualization, B.F.F. and W.H.F.; methodology, B.F.F. and W.H.F.; software, B.F.F. and W.H.F.; validation, B.F.F. and W.H.F.; formal analysis, B.F.F. and W.H.F.; investigation, B.F.F. and W.H.F.; resources, B.F.F. and W.H.F.; data curation, B.F.F. and W.H.F.; writing—original draft preparation, B.F.F. and W.H.F.; writing—review and editing, B.F.F. and W.H.F.; visualization, B.F.F. and W.H.F. All authors have read and agreed to the published version of the manuscript.

**Funding:** This research received no external funding.

**Institutional Review Board Statement:** Not applicable.

**Informed Consent Statement:** Not applicable.

**Data Availability Statement:** Data is contained within the article and Supplementary Material. Empirical data and the associated R code are available upon request.

**Conflicts of Interest:** The authors declare no conflict of interest.

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
