# Peer review of "Effect Sizes for Estimating Differential Item Functioning Influence at the Test Level"

_psych, doi:10.3390/psych5010013_

Round 1

Reviewer 1 Report

Dear Authors,

thank you very much for your study. You will find my comments below and in the attachment.

Kind Regards,

Comments

Point 1: General observation: The study can have a significant contribution on DTF and effect size estimation however I think that the authors should restructure the Introduction, and some part of the results and some part of the material and method section in a resubmission. My comments were as follows.

Point 2: Abstract: In my opinion, the abstract should contain more about the study. From L.4 to L.9 it says some general sentences about the DIF/effect size. These sentences could be better in the introduction. I would like to see a summary of the study in the abstract.

Point 3. Keywords: I am not familiar with the keyword "Scoring Inference", I would maybe say "Measurement Invariance ?".

Point 4. Introduction: I think the motivation for the research and its possible contribution to the field needs to be reconstructed with a more detailed literature review. The authors should better restructure why they need to compare the effect size methods with a simulation study and an applied data example.

Point 5:  L. 34 (sDTF) and unsigned (uDTF); DTF is the abbreviation for Differential Test Functioning, but I haven't read anything about it until this part. The authors explain DIF and DBF, but there is no information about DTF. The idea is to check whether the accumulation of DIF items at the test level is detectable or meaningful or not. I could not be sure with introduction part that accumulated amount of differential item functioning (DIF) in an item cluster/bundle (DBF) or accumulation of DIF items at the test level (DTF) was the focus of the study. For this reason, I would like to include in the introduction something about the DTF also.

Point 6:  L.88 “logistic regression (LR), a popular and accurate method for DIF detection”: accurate method should remove from this sentences, did Authors use it for DIF detection I did not see it in Material Methods?

Point7. Materials and Methods L.211 “Data generation and analysis”: I am not satisfied with the study design. In particular, there should be some guidance included here for researchers who would like to apply similar effect size methods to their studies.

1. Did the author generate the Rasch item difficulty parameter values themselves or did they take them from previous studies (if so, from which ones or if they generate, how they did.)

2. How did the Authors calculate the effect size statistics,( e.g., for sDTF and uDTF statistics, the "mirt" package can be used, or did the authors write their own codes for the approaches of Doebler (2019) and DTF statistics (if so, these may be shared in the supplemental files )

3.DIF detection, the authors should short explain in the study design which methods were used for DIF detection (Is it Raju`s area? L.232, L.233) and how they performed the analysis especially for real data sets.

Point 8:  L.222 “Prior research has shown that the assessment of DIF/DTF can be influenced by the test length: some examples of these studies should be added

Point 9:  L.224 “two assessment lengths”, The Authors has used both of test length and assessment length, only test length could be better

Point 10: Results:

a)     The authors were explained in the study design that different sample sizes (250/250, 500/500, 1000/1000, 2000/2000, 500/250, 1000/250, 2000/250, 1000/500, 2000/500, 2000/1000) and test lengths (10, 20 items) were manipulated. However, I did not see the results of these test conditions in Tables and these results must be reported with DIF type, magnitude of DIF, and proportion of DIF. For example, if the sample size of the groups is the same (250/250) and the test length is 10, how is the effect size statistic performed?

b)     Similarly, in abstract and in introduction the authors said that they applied in study also a real data set (L.11; L.46), however in material and methods, results sections I could not find about it. Which kind of the real data set has been used in study?

Point 11:  Table 4. Table 5 and Table 6: Some parts of the tables are in bold, is there a reason for this? If so, this should be explained in a small note.

Point 12:  L373 “Table7” There is not Table 7 in the Result section. Probably the Authors tried to say Table 6?

Author Response

Reviewer 1

Comments and Suggestions for Authors

Dear Authors,

thank you very much for your study. You will find my comments below and in the attachment.

Kind Regards,

Comments

Point 1: General observation: The study can have a significant contribution on DTF and effect size estimation however I think that the authors should restructure the Introduction, and some part of the results and some part of the material and method section in a resubmission. My comments were as follows.

Response

We appreciate the reviewer’s comment and agree that the introduction should be restructured and more details provided in the methods and results.  We have attempted to address each of the points raised by the reviewer, and found them to be helpful in improving the manuscript.

Point 2: Abstract: In my opinion, the abstract should contain more about the study. From L.4 to L.9 it says some general sentences about the DIF/effect size. These sentences could be better in the introduction. I would like to see a summary of the study in the abstract.

Response

We appreciate the reviewer’s suggestion and have rewritten the abstract with more specific discussion of our study.  The edited abstract appears below.

Differential item functioning (DIF) is a critical step in providing evidence to support a scoring inference in building a validity argument for a psychological or educational assessment. Effect sizes can assist in understanding the accumulation of DIF at the test score level. The current simulation study investigated the performance of several proposed effect size measures under a variety of conditions.  Conditions under study included varied sample sizes, DIF effect sizes, proportion of items with DIF, and type of DIF (additive vs nonadditive).  DIF effect sizes under study included sDTF%, uDTF%,  , , , , and .  The results of the study suggest that across study conditions , , and  were consistently the most accurate measures of the DIF effects. The effect sizes were also estimated in an empirical example. Recommendations and implications for practice are discussed.

Point 3. Keywords: I am not familiar with the keyword "Scoring Inference", I would maybe say "Measurement Invariance ?".

Response

We agree with the reviewer that “Measurement Invariance” would be preferable to “Scoring Inference”, and have made this change to the keywords in the manuscript.

Point 4. Introduction: I think the motivation for the research and its possible contribution to the field needs to be reconstructed with a more detailed literature review. The authors should better restructure why they need to compare the effect size methods with a simulation study and an applied data example.

Response

 Thank you for this point. We have added to the introduction the motivation for the simulation study and the applied example to show how they may work in applied situations. We have added references to past work that has compared a few effect sizes in this area and where applied data has been used to compare effect sizes.  This additional text appears on pages 3 and 4 of the revision, in the section preceding Effect size methods for quantifying the magnitude of DIF in an assessment.

Point 5:  L. 34 (sDTF) and unsigned (uDTF); DTF is the abbreviation for Differential Test Functioning, but I haven't read anything about it until this part. The authors explain DIF and DBF, but there is no information about DTF. The idea is to check whether the accumulation of DIF items at the test level is detectable or meaningful or not. I could not be sure with introduction part that accumulated amount of differential item functioning (DIF) in an item cluster/bundle (DBF) or accumulation of DIF items at the test level (DTF) was the focus of the study. For this reason, I would like to include in the introduction something about the DTF also.

Response

We agree with the reviewer that differential test functioning (DTF) was not properly introduced in the manuscript.  Therefore, we have included the following sentence on page 3 of the revision.  This is the same section in which DIF and DBF are first introduced.

When differential functioning is considered for the entire set of test items, we refer to differential test functioning (DTF). DTF has been investigated in different context from personality selection (Stark et al., 2004) to health care (Badia et al., 2002), including comparisons of effect sizes (Stark et al.).  Capturing a lack of item invariance at the test level can assist in assessing if such issues influence score level decisions and assisting with understanding the practical importance of a lack on invariance (Meade, 2010; Nye & Drasgow, 2011).

Point 6:  L.88 “logistic regression (LR), a popular and accurate method for DIF detection”: accurate method should remove from this sentences, did Authors use it for DIF detection I did not see it in Material Methods?

Response

We agree with the reviewer and have removed “accurate” from the sentence.  We did use the R-squared effect size from logistic regression in our study.  We have explicitly stated that the  effect size was obtained from logistic regression.  We hope that this clarifies the use of LR in our study.  If not, we will be happy to provide further explanation of this issue.

Point7. Materials and Methods L.211 “Data generation and analysis”: I am not satisfied with the study design. In particular, there should be some guidance included here for researchers who would like to apply similar effect size methods to their studies.

  1. Did the author generate the Rasch item difficulty parameter values themselves or did they take them from previous studies (if so, from which ones or if they generate, how they did.)
  2. How did the Authors calculate the effect size statistics,( e.g., for sDTF and uDTF statistics, the "mirt" package can be used, or did the authors write their own codes for the approaches of Doebler (2019) and DTF statistics (if so, these may be shared in the supplemental files )

3.DIF detection, the authors should short explain in the study design which methods were used for DIF detection (Is it Raju`s area? L.232, L.233) and how they performed the analysis especially for real data sets.

Response

We appreciate the reviewer’s comments and agree that more clarity needs to be provided regarding the simulation study methods.  Therefore, we have made the following changes to the manuscript.  We hope that these edits address the reviewer’s concerns.  If more changes are needed, we will be happy to make them.

  1. The item difficulty values were drawn from a sample of intelligence test score item responses. We have included the following sentence on page 11 of the revision to clarify this source.

The item difficulty parameters used to generate data for the reference group were drawn from a widely used intelligence test (Wechsler, 2008) and appear in Table 2.

  1. The effect sizes were all calculated with author created R code. We have included an empirical example with publicly available data and can make the data and code available to readers upon publication.

  1. We agree with the reviewer and have included an example with a real dataset that is publicly available. As noted above, the R code will also be made available to readers   This empirical example appears beginning on page 18 of the revision.

Point 8:  L.222 “Prior research has shown that the assessment of DIF/DTF can be influenced by the test length: some examples of these studies should be added

Response

We agree with the reviewer and on page 11 of the revision have included several citations to support this assertion.

Point 9:  L.224 “two assessment lengths”, The Authors has used both of test length and assessment length, only test length could be better

Response

We agree with the reviewer and have changed the text to read “test lengths” on page 11 of the revision.

Point 10: Results:

  1. a)The authors were explained in the study design that different sample sizes (250/250, 500/500, 1000/1000, 2000/2000, 500/250, 1000/250, 2000/250, 1000/500, 2000/500, 2000/1000) and test lengths (10, 20 items) were manipulated. However, I did not see the results of these test conditions in Tables and these results must be reported with DIF type, magnitude of DIF, and proportion of DIF. For example, if the sample size of the groups is the same (250/250) and the test length is 10, how is the effect size statistic performed?
  2. b)Similarly, in abstract and in introduction the authors said that they applied in study also a real data set (L.11; L.46), however in material and methods, results sections I could not find about it. Which kind of the real data set has been used in study?

Response

The ANOVA was used to identify the main effects and interactions that were significantly related to the study outcomes.  It is these results that are featured in the body of the manuscript.  However, we agree with the reviewer that including results for the main effects would be informative for readers.  Therefore, we have included these in supplemental tables at the end of the manuscript.

Point 11:  Table 4. Table 5 and Table 6: Some parts of the tables are in bold, is there a reason for this? If so, this should be explained in a small note.

Response

We appreciate the author’s noting this issue.  We did not intend to have any bold elements in the tables and have removed them in the revised manuscript.  Thank you for pointing this error out.

Point 12:  L373 “Table7” There is not Table 7 in the Result section. Probably the Authors tried to say Table 6?

Response

The reviewer is correct that this should be “Table 6”.  We have made the change in the manuscript and thank the reviewer for pointing out this error.

Reviewer 2 Report

This paper uses Monte Carlo simulation to compare several DIF effect size measures. Although this is a valuable area of study, the current paper appears to be somewhat unfocused and unclear about its implications. To illustrate, I mention particular issues below that I believe need to be addressed to improve the quality and impact of this work.

This paper would be much more valuable if we could compare estimated effect sizes to true (population) effect sizes. It's not obvious to me why these population values wouldn't be derive-able, but could the authors speak to why this approach was not taken? I can't see how you can make recommendations without knowing the accuracy relative to a population value. Similarly, taking the ratio of effect sizes in the DIF vs. no-DIF conditions is interesting, but it's not clear whether even these should be directly compared to each other. This ratio approach also seems to disregard accuracy and instead seems to assume that bigger effect sizes are better when DIF truly exists. 

I don't think that the paper mentions which items were simulated to have DIF - were they particular values from the items listed in Table 2 or randomly selected with each replication? Also, the title of Table 2 states that there are 30 items, but only 20 difficulties are listed. Which of these 20 items were included in the 10-item conditions?

It is somewhat difficult to understand the results since it does not seem that the different types of effect sizes should be directly compared to each other.

It is unclear how the results in the tables were analyzed, particularly how they were aggregated over 1000 replications. Specific examples include: the confidence intervals and standard errors in Table 3, standard errors in Table 4-6. I am also confused about how the 95% CIs relate to the SEs - e.g., in Table 3 sDTF% has a SE of 3.23 but the CI width is only .66.

The sDTF% function seems to be particularly bad at quantifying DIF. Is this because DIF effect sizes often "cancel out" when compared to each other? Particularly in the non-additive condition, might the null hypothesis actually be true for sDTF (which explains the low proportion outside the CI).

Unfortunately, this study appears to ignore its stated goals to "investigate existing 198 interpretive guidelines where available, and suggest tentative guidelines" (section 2.1). Instead, the focus on interpreting the proportion of CIs that include 0 turns the focus away from effect sizes (the nominal purpose of this article) and back toward hypothesis testing.

Aside from the above points, generating data from the Rasch model with relatively large magnitudes of DIF is a rather idealistic framework in which to study DIF. I would like to see the authors at least acknowledge this limitation, but ideally design a more realistic set of study conditions.

There are several formatting problems throughout this paper (e.g., equation numbering on page 3)

Author Response

Reviewer 2

Comments and Suggestions for Authors

This paper uses Monte Carlo simulation to compare several DIF effect size measures. Although this is a valuable area of study, the current paper appears to be somewhat unfocused and unclear about its implications. To illustrate, I mention particular issues below that I believe need to be addressed to improve the quality and impact of this work.

  1. This paper would be much more valuable if we could compare estimated effect sizes to true (population) effect sizes. It's not obvious to me why these population values wouldn't be derive-able, but could the authors speak to why this approach was not taken? I can't see how you can make recommendations without knowing the accuracy relative to a population value. Similarly, taking the ratio of effect sizes in the DIF vs. no-DIF conditions is interesting, but it's not clear whether even these should be directly compared to each other. This ratio approach also seems to disregard accuracy and instead seems to assume that bigger effect sizes are better when DIF truly exists. 

Response

Thank you for this point. We agree that the ratios could be confusing to interpret. We have removed this from the manuscript. We did spend time trying to derive analytically the population solution for all effect sizes. We were not confident in solving these for all effect sizes at this time. Thus, we took a different approach to address the spirit of your approach. We obtained the population parameter effect size values by generating populations consisting of 20,000,000 million individuals in each simulated group under a specific DIF condition (e.g., 0.8 magnitude with 20% of items containing DIF). To ensure the stability of these values, this procedure was conducted 3 times for each condition and it was noted that the resulting effect size values were virtually identical across these replications. These were used to represent population effect sizes. We hope this address your point.  

  1. I don't think that the paper mentions which items were simulated to have DIF - were they particular values from the items listed in Table 2 or randomly selected with each replication? Also, the title of Table 2 states that there are 30 items, but only 20 difficulties are listed. Which of these 20 items were included in the 10-item conditions?

Response

The reviewer raises an important point.  The items selected to have DIF were 2, 3, 6, and 7.  We have included the following sentence on page 12 of the revision to make this clear.

When DIF was present, the magnitude values listed above were added to the values for items 2, 3, 6, and 7 (for the appropriate proportion of items with DIF) in Table 2 for the target items in order to obtain item difficulty parameters for the focal group. 

  1. It is somewhat difficult to understand the results since it does not seem that the different types of effect sizes should be directly compared to each other.

Response

Thank you for this comment. The new results contain relative bias. This allows for comparing the performance of the effect sizes. We believe that this improves what can be said about the effect sizes measures relative to one another, when compared to the earlier version of the manuscript. We hope that this shift in focus improves the quality of the manuscript.

  1. It is unclear how the results in the tables were analyzed, particularly how they were aggregated over 1000 replications. Specific examples include: the confidence intervals and standard errors in Table 3, standard errors in Table 4-6. I am also confused about how the 95% CIs relate to the SEs - e.g., in Table 3 sDTF% has a SE of 3.23 but the CI width is only .66.

Response

We appreciate the reviewer’s comments and agree that more clarification regarding how the results were aggregated across the replications is required.  Essentially, for each combination of manipulated study conditions we calculated the standard deviation for each effect size across the 1000 replications.  This served as the empirical standard error that is described in the manuscript.  In the same way, we calculated the mean of each effect size statistic across the 1000 replications for each combination of study conditions.  We have added the following text to page 13 of the revised manuscript.  We hope that this clarifies how the results were aggregated in the study.  If, however, questions remain, we will be happy to address this issue with further revisions.

The empirical standard error is the standard deviation of the 1000 parameter estimates for each combination of conditions.  In other words, for each combination of the study conditions, the standard deviation of each effect size measure was calculated for the 1000 estimated values.  Likewise, the mean effect size value for each effect size statistic was calculated across the 1000 parameter estimates for each combination of conditions.   

We have removed the information about the 95% CIs.  They were in reference to the CIs for the no DIF condition, but we recognize that their use in this context was confusing and have thus taken them out of the revised manuscript.

  1. The sDTF% function seems to be particularly bad at quantifying DIF. Is this because DIF effect sizes often "cancel out" when compared to each other? Particularly in the non-additive condition, might the null hypothesis actually be true for sDTF (which explains the low proportion outside the CI).

Response

As noted above, based on reviewer feedback, we have refocused our study on the relative bias of the effect size estimates. 

  1. Unfortunately, this study appears to ignore its stated goals to "investigate existing 198 interpretive guidelines where available, and suggest tentative guidelines" (section 2.1). Instead, the focus on interpreting the proportion of CIs that include 0 turns the focus away from effect sizes (the nominal purpose of this article) and back toward hypothesis testing.

Aside from the above points, generating data from the Rasch model with relatively large magnitudes of DIF is a rather idealistic framework in which to study DIF. I would like to see the authors at least acknowledge this limitation, but ideally design a more realistic set of study conditions.

Response

We agree with the reviewer and have thus changed the focus of our study so that relative estimation bias and empirical standard errors are the outcomes of interest, and the CIs were removed.  We appreciate the reviewer’s suggestions and believe that incorporating them in the manuscript has greatly improved the quality of our work, particularly by keeping the focus on the effect sizes and not on hypothesis testing.  We hope that the reviewers and editor agree.

We agree that more work examining other modeling and DIF conditions is necessary, and therefore we have included recommendations to do so in the Directions for Future Research section of the revision.

  1. There are several formatting problems throughout this paper (e.g., equation numbering on page 3)

Response

Thank you for the comment.  We have gone through the manuscript and attempted to correct the formatting issues.  If we missed any, we will be happy to fix them.

Reviewer 3 Report

Thank you for allowing me to review your very interesting and useful paper.  I found many points of strength in your paper. First, it is a very well-written paper that can be easily read by a large audience. Second, you guide the reader throughout the paper supporting them in the comprehension of formulas, procedures, and testing. very nice. Last but not least, the content of the papers is intriguing and useful not only for psychologists but also for educational scientists and closer fields of study. So, by my side, congrats on your publication:)

Author Response

Reviewer 3

Comments and Suggestions for Authors

Thank you for allowing me to review your very interesting and useful paper.  I found many points of strength in your paper. First, it is a very well-written paper that can be easily read by a large audience. Second, you guide the reader throughout the paper supporting them in the comprehension of formulas, procedures, and testing. very nice. Last but not least, the content of the papers is intriguing and useful not only for psychologists but also for educational scientists and closer fields of study. So, by my side, congrats on your publication:)

Response

We appreciate the reviewer’s kind words very much.  Thank you.

Round 2

Reviewer 1 Report

Dear Authors,

Thank you very much for your reply. I am satisfied with the answers, however, I could not see on the website either the revised manuscript or the supplementary file to which you apply the changes. There is a revised manuscript, but it is not the final version, I think. Because the comments you agree with are not in the final manuscript either.

As result, I could not see the revised version with the changes. Could you please upload the revised version and the supplementary file to the website as well?

Kind Regards,

Author Response

* I don't think that it's necessary to use both terms DTF and DBF, since the distinction between the two doesn't impact any of your findings. Although it could be appropriate to discuss both concepts in the introduction, I don't think both abbreviations are necessary as the two sets of terminology could cause confusion.

 Response

We agree with the reviewer and have used DTF throughout the manuscript, with the exception of in the first paragraph where we describe DIF, DBF, and DTF.

* Please clarify how R^2 is calculated for logistic regression

Response

We appreciate this comment and have included an explanation, starting at line 168, of how R^2 delta is calculated for logistic regression. 

* Please give a rationale as to why items 2, 3, 6, and 7 were selected to exhibit DIF in the simulations. Would any of your results differ if you selected a different set of DIF items? Please also discuss how these are easier items than many in the test. 

Response

We agree that this issue should be clarified in the manuscript.  The items were selected randomly, and we do agree that results might differ had the more difficult items been selected.  We have clarified at line 340 of the revision that the items were randomly selected.  On page 27 (line 571), we have included the following sentence noting that these items were relatively easy and that future research should examine the performance of these effect sizes with more difficult items.

The items selected to have DIF in the current study tended to be relatively easy, with the most difficult having a population difficulty parameter value of 0.41.  Further simulations with more difficult items exhibiting DIF would be useful for future study.

* Please clarify that the supplemental tables (hopefully) present averages over Monte Carlo replications.

Response

Yes, these results are averaged across Monte Carlo replications.  We have added a statement to this effect to each supplemental table.

* Overall, I find the RB results more compelling than some of the other results presented. The paper could also be strengthened by providing a clearer rationale for how to understand the other metrics (esp. which comparisons are and are not appropriate, as the different measures are on different scales).

Response

We agree with the reviewer that the RB is the most useful outcome statistic, particularly for comparisons across effect size statistics.  The other statistics are most useful for comparison within the same effect size across simulation conditions.  We have added the following paragraph to page 14 (line 380) of the revision to make this point clear.

With respect to interpretation of the outcomes, the RB is most useful for comparison across the various effect size statistics because it accounts for differences in the scale of each statistic by standardizing to the population value.  In other words, for each effect size measure, RB reflects the same metric, difference between the estimated and population values divided by the population value.  The empirical standard error is most useful for comparisons within the same effect size statistic across different simulation conditions.  It is not helpful for comparing across effect size measures because they are calculated on different scales.  Finally, the mean of the raw effect size statistic is also useful for comparison across simulation conditions within effect size measure and not across effect sizes, again due to differences in scale.

Reviewer 2 Report

Thank you for this revision. I find the rationale and presentation of results much improved. I have only minor suggestions for polishing this manuscript:

* I don't think that it's necessary to use both terms DTF and DBF, since the distinction between the two doesn't impact any of your findings. Although it could be appropriate to discuss both concepts in the introduction, I don't think both abbreviations are necessary as the two sets of terminology could cause confusion.

* Please clarify how R^2 is calculated for logistic regression

* Please give a rationale as to why items 2, 3, 6, and 7 were selected to exhibit DIF in the simulations. Would any of your results differ if you selected a different set of DIF items? Please also discuss how these are easier items than many in the test. 

* Please clarify that the supplemental tables (hopefully) present averages over Monte Carlo replications.

* Overall, I find the RB results more compelling than some of the other results presented. The paper could also be strengthened by providing a clearer rationale for how to understand the other metrics (esp. which comparisons are and are not appropriate, as the different measures are on different scales).

Author Response

(The authors gave the same response as above.)
